# The B-cell inhibitory receptor CD22 is a major factor in host resistance to *Streptococcus pneumoniae* infection

**Vitor E. Fernandes**[1¤a]*, **Giuseppe Ercoli**[2], **Alan Bénard**[3], **Carolin Brandl**[4], **Hannah Fahnenstiel**[4], **Jennifer Müller-Winkler**[4], **Georg F. Weber**[3], **Paul Denny**[5¤b], **Lars Nitschke**[4☯]*, **Peter W. Andrew**[1☯]*

**1** Department of Infection, Immunity and Inflammation, University of Leicester, Leicester, United Kingdom, **2** Department of Genetics, University of Leicester, Leicester, United Kingdom, **3** Department of Surgery, University Hospital Erlangen, Erlangen, Germany, **4** Division of Genetics, Department of Biology, University of Erlangen, Erlangen, Germany, **5** Mammalian Genetics Unit, Medical Research Council, Harwell, United Kingdom

☯ These authors contributed equally to this work.
¤a Current address: Centre for Biomedical Research (CBMR), University of Algarve, Campus of Gambelas, Faro, Portugal.
¤b Current address: HUGO Gene Nomenclature Committee (HGNC), European Bioinformatics Institute (EMBL-EBI), European Molecular Biology Laboratory, Wellcome Genome Campus, Hinxton, Cambridgeshire, United Kingdom.
* vsfernandes@ualg.pt (VEF); lars.nitschke@fau.de (LN); pwa@le.ac.uk (PWA)

**Data Availability Statement:** All relevant data are within the manuscript and its Supporting Information files.

## Abstract

*Streptococcus pneumoniae* is a major human pathogen, causing pneumonia and sepsis. Genetic components strongly influence host responses to pneumococcal infections, but the responsible loci are unknown. We have previously identified a locus on mouse chromosome 7 from a susceptible mouse strain, CBA/Ca, to be crucial for pneumococcal infection. Here we identify a responsible gene, *Cd22*, which carries a point mutation in the CBA/Ca strain, leading to loss of CD22 on B cells. CBA/Ca mice and gene-targeted CD22-deficient mice on a C57BL/6 background are both similarly susceptible to pneumococcal infection, as shown by bacterial replication in the lungs, high bacteremia and early death. After bacterial infections, CD22-deficient mice had strongly reduced B cell populations in the lung, including GM-CSF producing, IgM secreting innate response activator B cells, which are crucial for protection. This study provides striking evidence that CD22 is crucial for protection during invasive pneumococcal disease.

## Author summary

*Streptococcus pneumoniae* (known as the pneumococcus) is a human bacterial pathogen responsible for diseases such as pneumonia and sepsis, that cause illness and death in millions of individuals. Susceptibility to pneumococcal infections is associated with genetic components that strongly influence how infected individuals respond to infection, but little is known about the causal gene(s) and the mechanisms of control of the infection. In

**Funding:** Research by PWA was funded by a grant by the UK Medical Research Council (https://mrc.ukri.org) grant MC_U142684174. Research by LN was funded by the German Research Foundation DFG (https://www.dfg.de) through TRR130 (P04) and CRC1181 (B6). Research by GFW was funded by the German Research Foundation DFG, by grants WE4892/3-1 and WE4892/4-1. The funders had no role in study design, data collection and analysis, decision to publish, or preparation of the manuscript.

**Competing interests:** The authors have declared that no competing interests exist.

previous studies we have found strong differences in susceptibility and resistance to pneumococcal infections between mouse strains. In this study we identified a gene, the *Cd22* gene, that controls resistance to pneumococcal infection. Mice without the B-cell specific CD22 protein were much more susceptible to infection with *S. pneumoniae*. We could show that a protective population of B cells that migrates to the lung during pneumococcal infection is missing in *Cd22*-deficient mice. The study shows to a new role for CD22 and indicates a new potential target for treatment of pneumococcal infections.

## Introduction

*Streptococcus pneumoniae* is a major human pathogen responsible for a spectrum of diseases, including pneumonia and sepsis, causing illness and death in millions of individuals. Susceptibility to pneumococcal infections is associated with genetic components that strongly influence the host responses to infection, but little is known about the causal gene(s). Our groups previously used mouse models of respiratory disease as tools to dissect the genetic factors that influence the host immune responses to invasive pneumococcal disease [1]. These previous studies found a clear spectrum of susceptibility and resistance to pneumococcal infections in inbred mouse strains. While resistant BALB/c mice showed a median survival time of > 168 hours after intranasal infection, and some other strains such as C57BL/6 or DBA/2 had intermediate survival times of 70 to 85 hours, the CBA/Ca strain was highly susceptible, with a survival time of only 27 hours after intranasal infection with *S. pneumoniae* [1]. The high susceptibility of CBA/Ca mice was associated with a high bacteremia, whereas no bacteria were detected in the blood of resistant BALB/c mice.

In order to map the responsible genetic locus for susceptibility or resistance to *S. pneumoniae* infection, intercrosses of the BALB/c and CBA/Ca strains were done and F2 mice were used to map a major locus controlling the development of bacteremia and survival after intranasal infection with *S. pneumoniae*. We identified this major locus on chromosome (Chr) 7, near D7Mit77, which showed high logarithm of the odds (LOD) scores associated with survival time (LOD = 6.3) and bacteremia (LOD = 4.6) [2]. The quantitative trait locus (QTL) on Chr. 7 was named *S. pneumoniae infection resistant 1* (*Spir1*). This locus identified on Chr 7 covered 11 Mb and contained about 250 genes.

We also performed genome-wide association studies (GWAS) to map genetic loci associated with susceptibility to pneumococcal infections in 26 inbred mouse strains. Four candidate QTLs were identified, on Chr. 7, 13, 18 and 19 [3]. Notably, the QTL on Chr. 7 was located within the *Spir1* locus, previously identified by the linkage study in BALB/c x CBA/Ca F2 mice. Within the hundreds of the genes in the *Spir1* QTL, only 22 genes showed phenotype-associated polymorphisms.

Now we report that a natural mutation in the *Cd22* gene within *Spir1* is the major explanation of the susceptibility of CBA/Ca mice to intranasal *S. pneumoniae* infection. CD22 (Siglec-2) is a B-lymphocyte-specific receptor and negatively regulates B cell receptor signaling. CD22 is implicated in B cell survival and proliferation and in induction of B cell tolerance and control of susceptibility to autoimmune diseases [4–6]. We found that the natural mutation in the *Cd22* gene of CBA/Ca mice causes a stop codon in the first immunoglobulin-like domain of CD22, leading to a non-functional protein. Also, CD22-deficient mice on a pure C57BL/6 background (CD22-/- mice) showed the same susceptibility. We demonstrated a strong reduction of B cells in the lungs of CBA/Ca and CD22-/- mice after pneumococcal lung infections, in contrast to innately resistant mice. Importantly, GM-CSF producing innate response

activator (IRA) B cells were strongly diminished in the lungs of mice lacking functional CD22. This shows an unexpected, and novel, role of the B cell protein CD22 in immunity to pneumococci.

## Results

### A polymorphism in the *Cd22* gene of CBA/Ca mice leads to lack of the CD22 protein

As observed previously (1) the mouse strain CBA/Ca shows a strong susceptibility to intranasal infections with *S. pneumoniae*. While all CBA/Ca mice died within 36 hours of intranasal infections, all BALB/c mice survived more than 168 hours (Fig 1A). This led to a strong disease score in CBA/Ca mice but no disease signs in BALB/c mice (Fig 1B). While the number of bacteria was controlled in the lungs of BALB/c mice after 24 hours, this did not occur in CBA/Ca mice (Fig 1C), coinciding with high bacteremia in the blood of CBA/Ca mice (Fig 1D). In order to determine the responsible locus, we sequenced the *Spir1* locus and within the locus observed 40 SNPs in the *Cd22* gene differing between BALB/c and CBA/Ca genomes. These were synonymous and non-synonymous mutations but, most notably, a SNP in the CBA/Ca *Cd22* gene at position 7:30,877,586 was predicted to introduce a stop codon (TAA). The tool, *Variant Effect Predictor*, provided at the *Ensembl* website (www.ensembl.org), was used to investigate variations between the sequences of 37 inbred mouse strains within the database. None of the 37 inbred strains in *Ensembl* had the SNP found in CBA/Ca (S1 Fig).

The SNP in the *Cd22* gene (C->T) of CBA/Ca, that results in a stop codon, is located in exon 4 of the *Cd22* gene, which encodes for the first CD22 Ig domain that contains the ligand-binding site. The SNP is located within the codon for amino acid 99 (Q) and the resulting stop codon is predicted to prevent synthesis of the bulk of the 130kd protein (Fig 2A). If the 98-amino acid N-terminus of the CD22 was synthesised, it was predicted that it would not be a stable product. CD22 has 7 extracellular Ig domains and its function is dependent on its membrane position [6]. To confirm the loss of the CD22 protein from B cells of CBA/Ca mice, splenic cells and lung cells were stained with anti-CD22 monoclonal antibody OX-97, before and after intranasal *S. pneumoniae* infection. While BALB/c mice expressed CD22 on their splenic and lung B cells, whether infected or not, CD22 was totally undetected on B cells from noninfected and infected CBA/Ca mice (Fig 2B). The lack of CD22 on B cells of CBA/Ca mice was confirmed with another CD22-specific monoclonal antibody, Cy34.1 (S2 Fig). Because both monoclonals bind to the second Ig-like domain of CD22 (d2 in Fig 2A) [7, 8], it can be concluded that CBA/Ca mice lack the CD22 protein on the B cell surface.

### CD22-deficient mice show a high susceptibility to *S. pneumoniae* infections

CBA/Ca mice, which lack CD22 protein, showed a high susceptibility to *S. pneumoniae* infections [1]. To determine whether this phenotype is directly caused by the loss of the CD22 protein, CD22-deficient mice on a pure C57BL/6 background and C57BL/6 control mice were infected with *S. pneumoniae* and monitored for 7 days.

After pneumococcal infection, very different survival rates were observed between C57BL/6 (CD22-positive) and mice with a CD22-deficiency on a C57BL/6 background. The survival rate of the CD22-/- group was much lower than the control group (p<0.001), with 100% of the CD22-deficient mice reaching the experimental endpoint by 72h post-infection, whereas 56% of the control mice survived to the end of the experiment, at 168h (Fig 3A). The median survival time of the CD22-deficient mice that survived was 45h, while the C57BL/6 that succumbed had longer survival time of 82h (p<0.001). Following the infection, the five C57BL/6

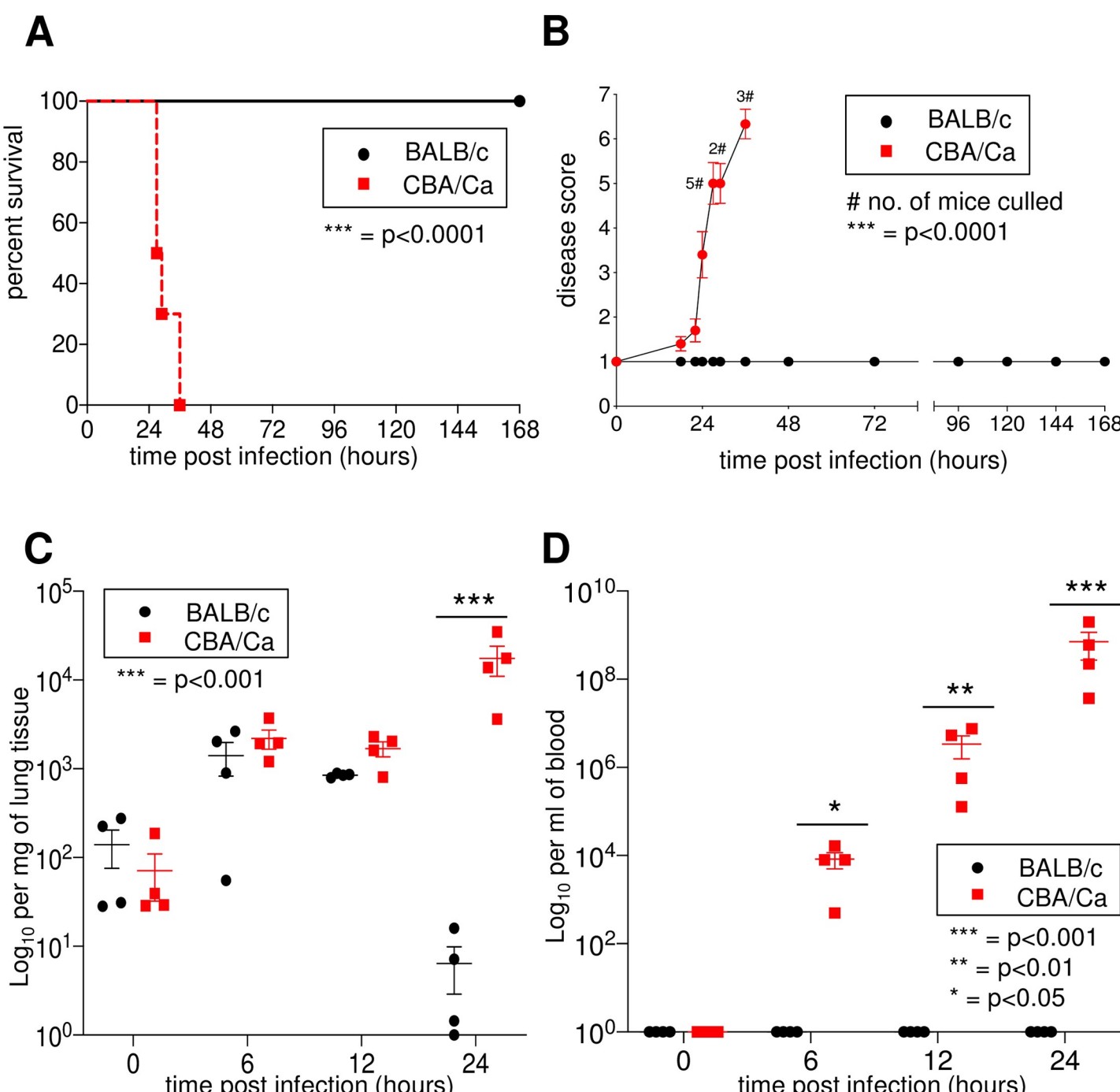

**Fig 1. CBA/Ca mice show a strong susceptibility after intranasal *S. pneumoniae* infections, when compared to BALB/c mice.** A) Survival of infected mice monitored for 7 days. B) Disease scores. C) Number of bacteria per mg of lung tissue or D) per ml of blood (D). Data represent mean values ± SEM. Statistics were done with Log-rank test (for A), two-tailed *t* test (for B) and two-way ANOVA followed by Bonferroni post-test (for C, D). * = p<0.05, ** = p<0.01, *** = p<0.001. A) and B): 10 animals per group; C) and D): more than 4 animals per group. Results are representative of 2 independent experiments.

mice that survived showed only very mild disease signs. The increased susceptibility of the CD22-deficient mice was reflected in the early appearance of signs of disease and the rapid progression of the disease score (Fig 3B).

## A

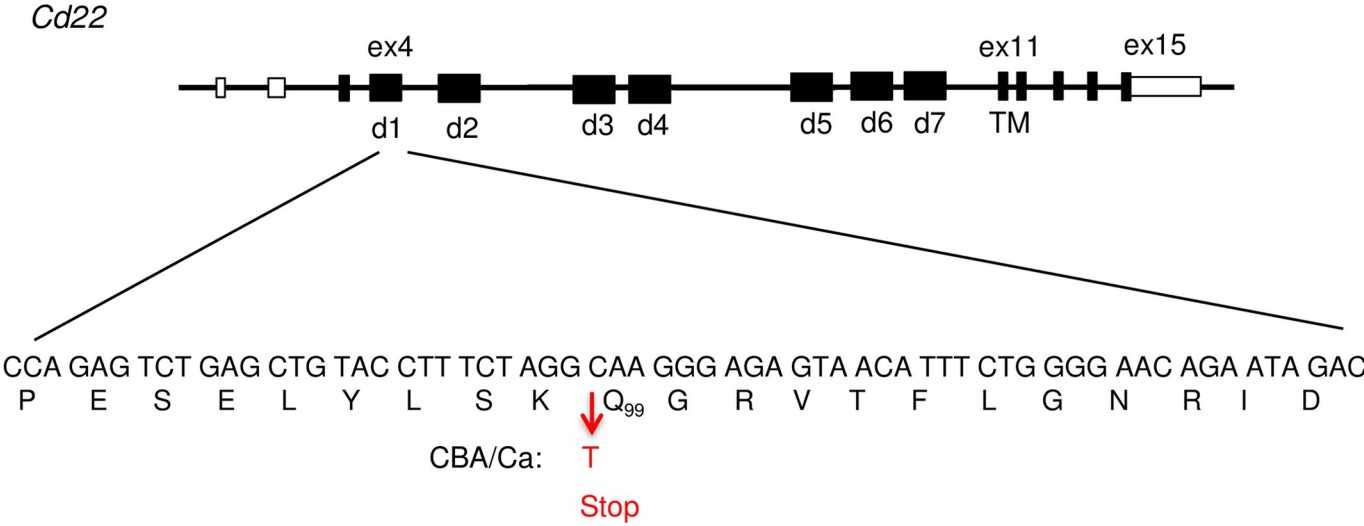

## B

SPLEEN — LUNG

BALB/c — CBA/Ca — BALB/c — CBA/Ca

Uninfected

Infected

CD22

CD19

**Fig 2. CBA/Ca mice carry a mutation in *Cd22* which leads to lack of CD22 expression on B cells.** A) Structure of the *Cd22* gene. The highlighted DNA and amino acid sequence of exon 4 shows the sequence of most mouse strains and the mutation found in CBA/Ca mice (C->T) causing an introduction of a stop codon. B) Flow cytometric analysis of splenocytes and lung cells from BALB/c and CBA/Ca mice stained for CD19 and CD22 (Mab OX-97) in the absence of infection and after intranasal infection with *S. pneumoniae*.

Next, the progress of pneumococcal disease in CD22-/- mice and controls was investigated in an experiment that determined pneumococcal numbers in the lung, blood and spleen at over 48 hours post-infection. Although there was a tendency of higher CFUs in the lungs of CD22-/- mice, there were no significant differences (p>0.05) in the number of bacteria in the lungs of CD22-/- mice and the control group, at any of the time points (Fig 3C). In contrast, there was a strong dissemination of bacteria to the blood (Fig 3D) and spleen (Fig 3E) of CD22-/- mice. The numbers of bacteria progressively increased in the blood and spleen between 12 and 48 hours in CD22-/- mice. In contrast, hardly any bacteria were detected either in the blood or spleen of the C57BL/6 (CD22+/+) mice (Fig 3D and 3E).

## After *S. pneumoniae* infection, CD22-deficient mice have impaired B cell populations in the lung

After intranasal infection with *S. pneumoniae*, CBA/Ca mice showed impaired B cell numbers in the lung, when compared to the BALB/c strain of mice (S3A Fig). In the spleen the two strains show no difference in B cell numbers up to 12 hours after infection, but at 24h after the infection higher B cell numbers were detected in CBA/Ca (S3B Fig). The cellular composition of immune cells infiltrating the lungs was analysed in more detail after *S. pneumoniae* infection of CD22-/- and C57BL/6 (CD22+/+) mice (Fig 4). Numbers of neutrophils (Fig 4A) and eosinophils (Fig 4B) increased over time after infection, but there were no consistent differences between the two mouse strains. T cells and alveolar macrophages increased in numbers and there were also no consistent differences between CD22-/- and C57BL/6 mice (Fig 4C and 4D). However, numbers of conventional B cells, as well as B1a cells were highly elevated in the lungs of C57BL/6 mice at each time point following infection, but much less so in CD22-/- mice (Fig 4E and 4F). Consequently, numbers of both conventional B2 cells (Fig 4E), as well as of B1a cells (Fig 4F), were significantly higher in the C57BL/6 lungs compared to the CD22-/- at all time points. In contrast, B2 and B1a cell numbers were unchanged in the spleen of both strains after infection, with the exception of lower B2 cells in CD22-/- mice at 48 hours (Fig 4G and 4H). From earlier studies it is known that there are no differences in B cell numbers in uninfected CD22-/- and CD22+/+ (C57BL/6) mice in the spleen, lymph nodes or blood [9–12].

We examined whether impaired B cell homing to the lung or impaired cellular proliferation/ survival in the lung is the mechanism responsible for the B-cell defect after *S. pneumoniae* infections. For this purpose, lung homing assays after cellular transfers with WT and CD22-/- B cells were performed. Splenic cells of C57BL/6 or CD22-/- mice were labelled with the dyes CFSE or CTV, respectively, then mixed in a 1:1 ratio and co-injected i.v. into C57BL/6 recipient mice (Fig 5A). In order to avoid an influence of the dyes, in some experiments the dyes were also swapped between the splenocytes of the two strains, without any change of results. The recipient mice were analysed at an early time point (2h after cell transfer) and at two later time points (24h and 48h). While there was no difference between WT and CD22-/- B cell numbers detected in the lungs of recipients after 2 h or 24 h, there was a clear reduction of B cell numbers at 48 h (Fig 5B). In contrast, T cell numbers of both donors in the lungs were similar. Also no differences of B cell numbers of both donors in the spleen of recipient mice were observed (Fig 5B). This shows that the initial homing of CD22-/- B cells to the lungs is not

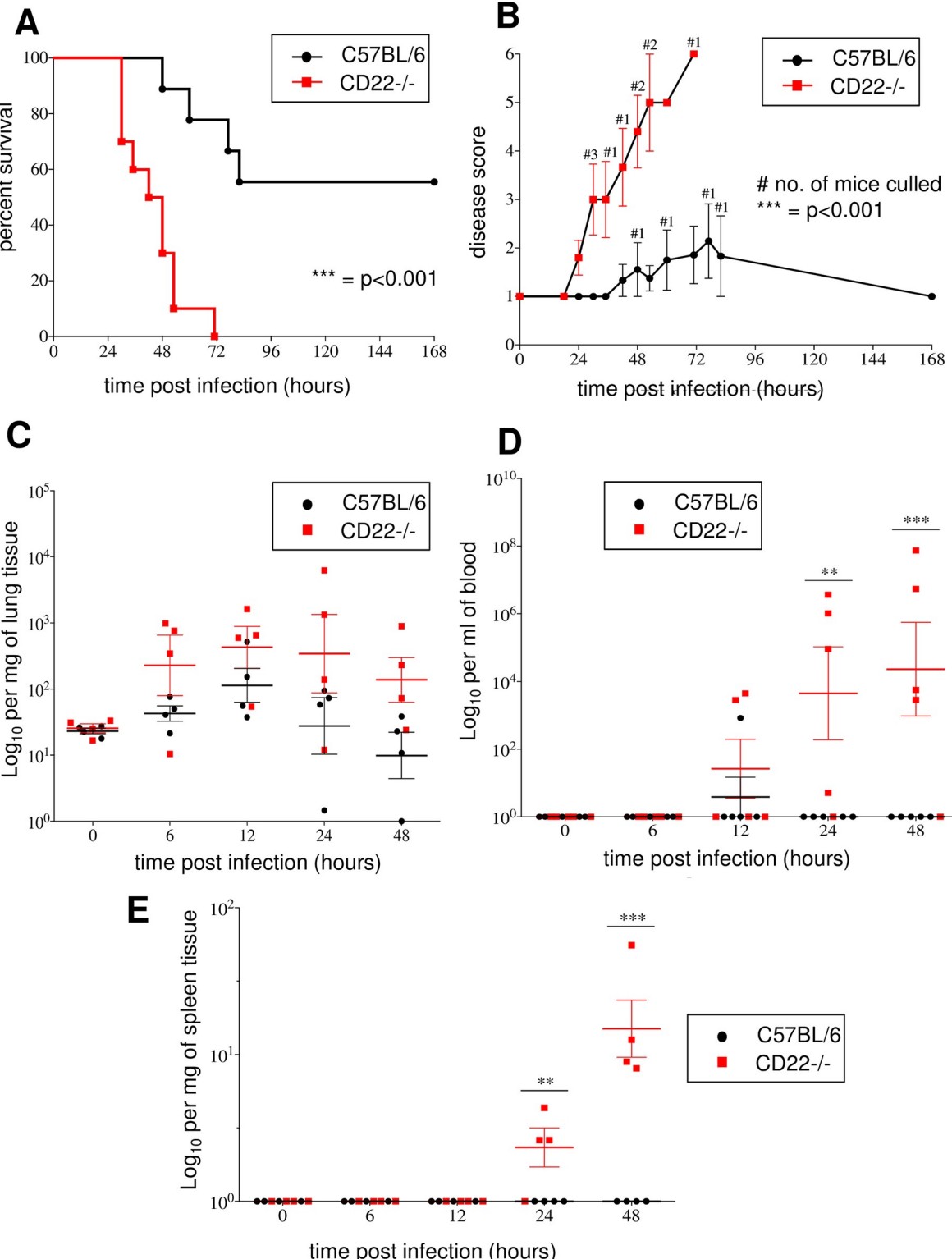

**Fig 3. Strong susceptibility of CD22-/- mice to *S. pneumoniae* infections, when compared to C57BL/6 (CD22+/+) mice.** A) Survival of infected mice monitored for 7 days. B) Disease scores. Number of bacteria per mg of lung tissue (C), per ml of blood (D) or per mg of spleen tissue (E). Data represent mean values ± SEM. Statistics were done with Log-rank test (for A), two-tailed *t* test (for B) and two-way ANOVA followed by Bonferroni post-test (for D, E). * = p<0.05, ** = p<0.01, *** = p<0.001. A) and B): 9–10 animals per group; C-E: 5 animals per group. Significant increase of number of pneumococci in the lung tissue of CD22-/- and control mice (Figure C), between 0

and 12 hours p.i. (p<0.05); significant increase of CFU numbers in the blood of CD22-/- mice (p<0.01) (Figure D); and significant increase of CFU numbers in the spleen of CBA/Ca mice (p<0.001). Results are representative of 2 independent experiments.

impaired, however CD22-/- B cells seem to show less retention in the lung. By comparison of the dilution of CFSE or CTV, respectively, we did not observe different proliferation of WT and CD22-/- B cells. We therefore suggest that the reduced CD22-/- B cell numbers in the lung at 48h may indicate an impaired survival.

## CD22-deficient mice have an impaired population of GM-CSF producing IRA B cells in the lung, both after *S. pneumoniae* or *E. coli* infections

In order to further examine the mechanism for the high susceptibility of CD22-/- mice to pneumococcal infection, secreted IgM and GM-CSF was measured in the bronchoalveolar lavage (BAL) of infected mice, because natural IgM that is produced by GM-CSF secreting B1a–like cells has been shown to be crucial for protective anti-bacterial responses in the lung [13]. We found lower IgM levels in the BAL of CD22-/- mice compared to the C57BL/6 control at early time points (3 hours and 6 hours) (p<0.01) after infection (Fig 6A). Also, lower GM-CSF levels were detected 3 hours after infection in the BAL of CD22-/- mice compared to the control (p<0.05) (Fig 6C), whereas IgM and GM-CSF concentrations in the blood were similar in CD22+/+ and CD22-/- mice (Fig 6B and 6D). IgM and IgG levels were also determined in CBA/Ca and BALB/c mice after pneumococcal infections. Here only a mild reduction of IgM in the BAL of CBA/Ca mice was detected at 24 hours after infection, while the IgG levels were similar in both types of mice (S4 Fig).

GM-CSF can be produced by several cell types [14], therefore we wanted to determine the source of GM-CSF produced by B cells in the lung. GM-CSF producing B cells (IRA B cells) are known to be recruited to the lung after bacterial infections [13]. To determine IRA B cells in the lung, C57BL/6 control and CD22-/- mice were infected with *S.pneumoniae* intranasally ($1 \times 10^5$ CFU, a lower than normal dose to obtain better survival) and analysed 72 hours after infection. At this time point IRA B cells in the lung (defined as CD19+, CD43+ CD93+) were determined. Fig 7A shows the flow cytometry data and gating strategy for lung IRA B cells. CD22-/- had significantly lower levels of IRA B cells in the lungs (p<0.05) (Fig 7B). Also, the number of GM-CSF+ IRA B cells was significantly lower (p<0.05), and there was at least a tendency to lower intracellular IgM+ IRA B cell numbers in the lungs CD22-/- mice compared to the C57BL/6 (Fig 7B). In contrast, this difference was not found in the spleen (Fig 7C).

IRA B cells were first described after *E. coli* infections of the lungs of mice [13]. In order to use the same model in which IRA B cells were first detected, CD22-/- and C57BL/6 control mice were intranasally infected with *E. coli*. Similar to pneumococcal infections, intranasal infection with *E. coli* resulted in a significantly higher mortality (p<0.001) and higher CFUs in CD22-/- mice (p<0.001) (Fig 8A and 8B). As we wanted to determine IRA B cells after *E.coli* infection with a protocol consistent with a previous report [13], we did intratracheal infections with *E.coli*. All CD22-/- mice and the C57BL/6 controls survived intratracheal infection for at least 72 hours post-infection with *E. coli*. 72 hours after intratracheal infection with *E. coli* CD22-/- mice had significantly lower levels of IRA B cells in the lungs (p<0.01) (Fig 8C). Also, the number of GM-CSF+ or intracellular IgM+ IRA B cells was significantly lower (p<0.01, p<0.05, respectively) in the lungs of CD22-/- mice compared to the C57BL/6 mice (Fig 8C). In the spleen, CD22-/- mice infected with *E. coli* also showed lower intracellular IgM+ IRA B cells, while the number of GM-CSF+ IRA B cells was not significantly different (Fig 8D). We conclude that both after *S.pneumoniae* and *E. coli* infections CD22-/- mice have an impaired

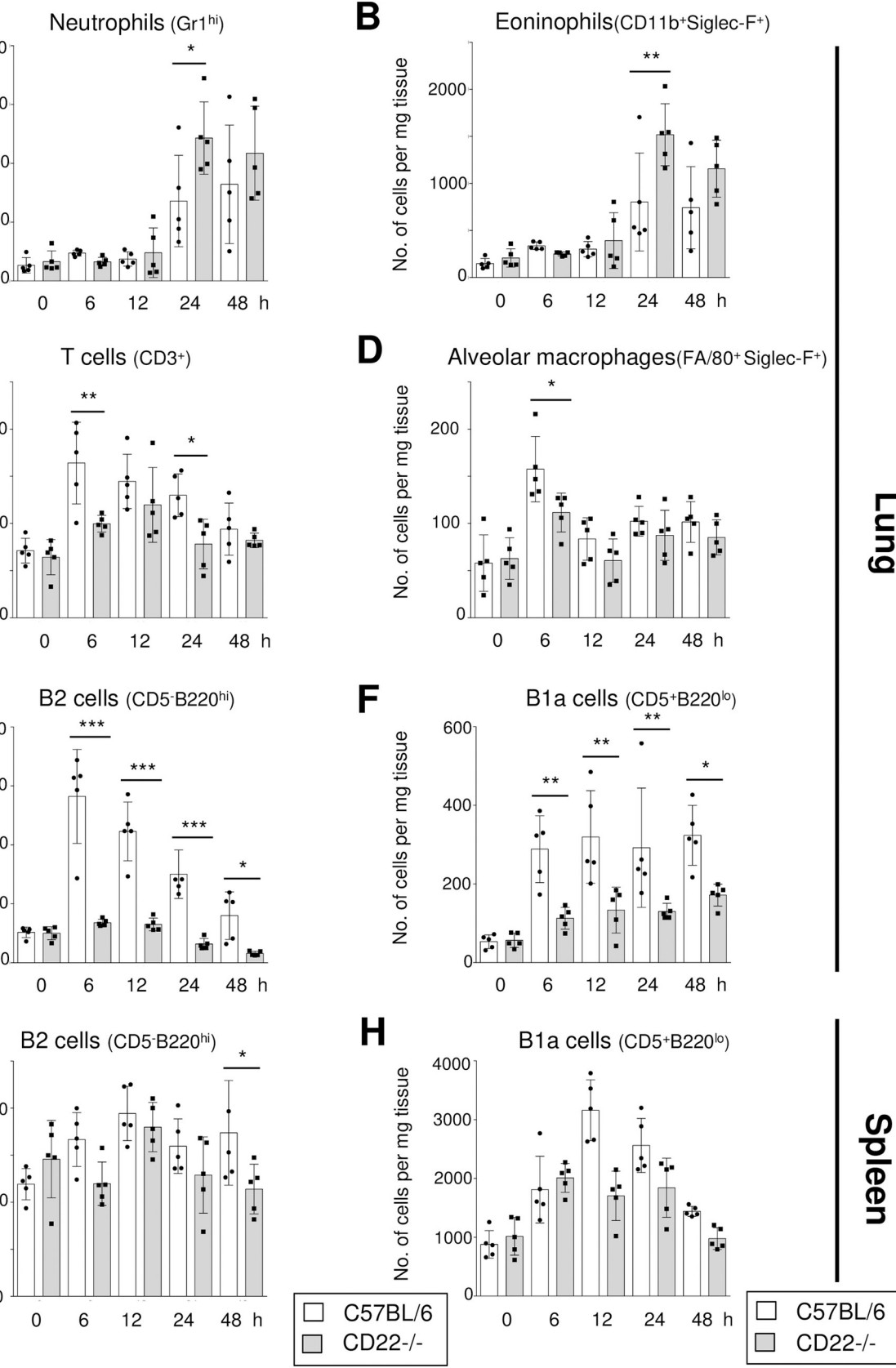

**Fig 4. CD22-deficient mice show impaired B cell numbers in the lung after *S. pneumoniae* infections.** Numbers of neutrophils, eosinophils, T cells, alveolar macrophages, B2 cells and B1a cells were measured in the lungs (A-F) and in the spleen (G,H) by flow cytometry at various time points up to 48 hours (h) after intranasal S. pneumoniae infection. Cell populations were determined with the shown markers. Significant differences between C57BL/6 and CD22-/- mice at each time point are shown. Data are means ± SEM. * = p<0.05, ** = p<0.01, *** = p<0.001. Significant increase over time of (A) neutrophil numbers observed in the lungs of control and CD22-deficient mice (p<0.01), as well as of (B) eosinophils in the lungs of control (p<0.001) and CD22-deficient mice (p<0.01), of (C) T cells in the lungs of control (p<0.01) and CD22-deficient mice (p<0.01), of (D) macrophages in the lungs of control (p<0.01) and CD22-deficient mice (p<0.05), of (E) B2 cells in the lungs of control (p<0.001) and CD22-deficient mice (p<0.001) and of (F) B1a cells in the lungs of control (p<0.01) and CD22-deficient mice (p<0.01).

IRA B cell population in the lung, with impaired GM-CSF and IgM production, and therefore lack an important mechanism for protection against bacterial infections.

## Discussion

Here we report the surprising finding that CD22, a member of the sialic acid binding Ig-like Siglec family of lectins, is an essential factor in innate resistance to pneumococcal disease. This is a surprising finding because CD22 has so far only been described as a negative regulator of B cell signalling [5, 6].

Within the B cell population, B-1 cells are known to play a critical role in response to pneumococcal infections. B-1 cells are innate-like B cells and the B-1a cell subpopulation produces phosphorylcholine- specific IgM, which serves as a first line defence against pneumococci [15, 16]. Upon bacterial lung infection of mice with pneumococci, B-1a cells from the peritoneal cavity can relocate to the lung where they produce protective IgM. It has been shown that a particular B cell subset producing the cytokine GM-CSF, called IRA B cells (for innate-response activator B cells) is crucial for this response [13]. Now we can conclude that CD22 is a critical factor for the contribution of B cells to the innate anti-pneumococcal responses.

The role of CD22 in immune responses previously has been studied with different classes of model antigens. Although thymus-independent antibody responses of CD22-deficient mice to antigens, such as NP-ficoll, are impaired due to missing or non-functional B cell subpopulations, such as marginal zone (MZ) or B-1b B cells [17, 18], thymus-dependent antibody responses to protein antigens are largely normal [9–12]. The loss of such a negative regulator would be anticipated to lead to stronger B cell responses. Consistent with this notion, CD22-deficient mice show signs of B cell hyperactivation and can develop autoimmune diseases, particularly when in combination with deficiency of another B-cell Siglec, Siglec-G [19, 20].

The role of CD22 in infections has not been widely studied but to be in line with the observations on autoimmunity [19, 20], it might be predicted that loss of CD22 would not have a negative impact on the outcome of acute infection. Hence, a single publication reported CD22$^{-/-}$mice have normal morbidity and mortality after staphylococcal infection [21]. In that study, injections with staphylococci were done intraveneously, which might explain the different results to intranasal pneumococcal infections reported here. Conversely, CD22 has been shown to play a role in controlling a viral infection. CD22-deficient mice are more susceptible to West Nile Virus (WNV) infection, after which there was normal anti-WNV antibody production but decreased WNV-specific CD8$^{+}$ T cell responses compared to wild type mice, including decreased lymphocyte migration into the draining lymph nodes [22].

It seems that a crucial B cell subpopulation is affected by the CD22-deficiency. After both pneumococcal and *E. coli* lung infections, CD22-deficient mice have a strongly reduced population of GM-CSF producing IgM secreting B cells in the lung. It has been shown, that in microbial airway infections, pleural B cells migrate to the lung where they produce protective IgM [15, 16]. GM-CSF producing IRA B cells are crucial for this process, as mice with a B-cell

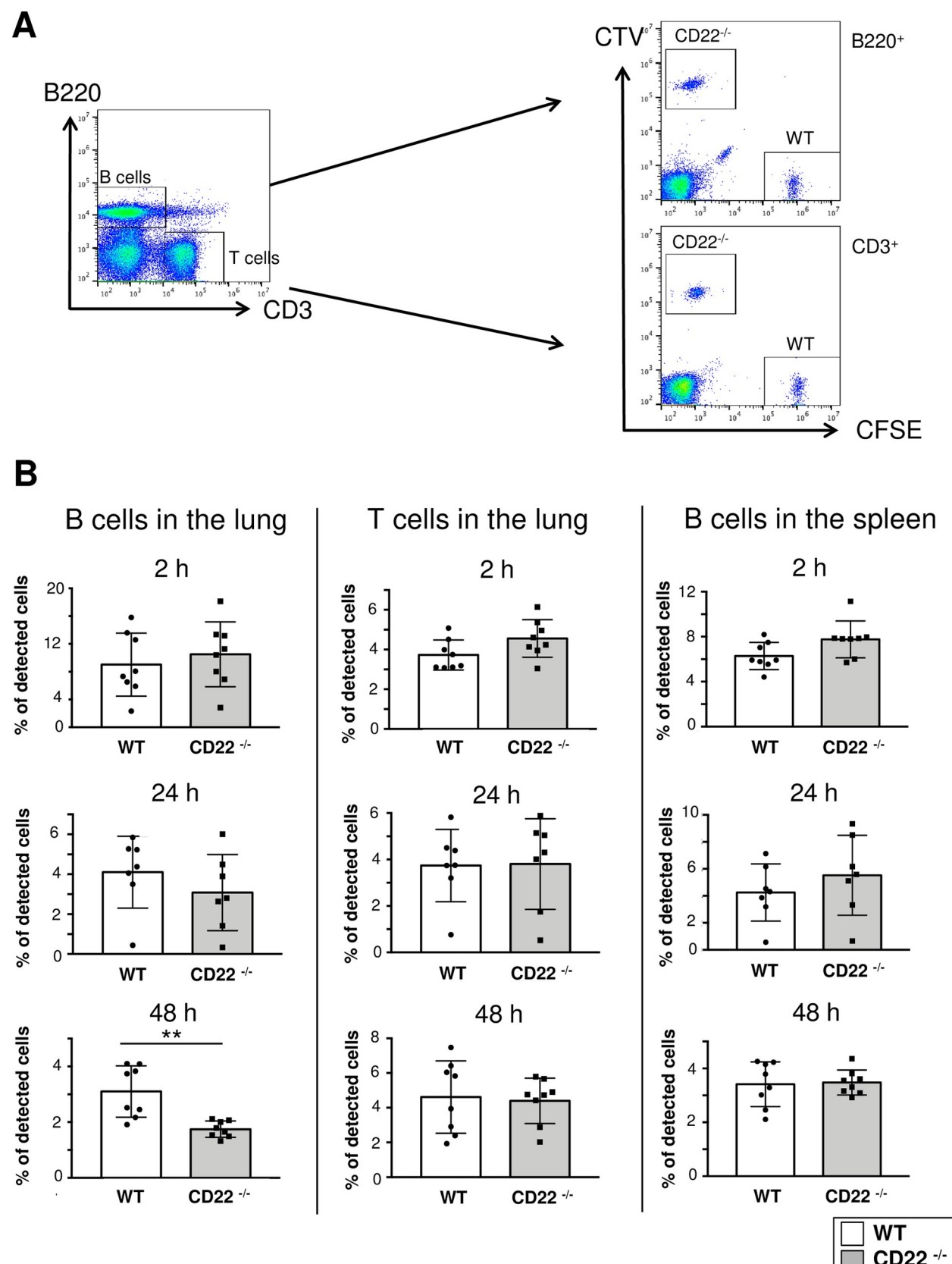

**Fig 5. Reduced CD22-/- B cell numbers in the lungs of recipient mice detected 48 hours after cellular transfer.** Flow cytometric analysis of lung and spleen cells from C57BL/6 mice injected i.v. with CFSE labelled C57BL/6 WT splenocytes and CTV labelled CD22$^{-/-}$ splenocytes (or vice versa) in a 1:1 mix. At various time points lungs and spleens were collected and stained for B220 and CD3. A) Gating strategy for the analysis of the proportion of WT and CD22$^{-/-}$ B and T cells found in the lung of recipient WT mice. B) Analysis of the percentage of WT and CD22$^{-/-}$ B and T cells found in the lung and spleen 2, 24 and 48 hours after i.v. injection. The proportion of WT and CD22$^{-/-}$ donor cells is displayed as the ratio of the percentage of CTV or CFSE-positive B or T cells found in the lung or spleen of recipient mice, divided by the percentage of CTV or CFSE-positive B or T cells injected (input) x 100. Bar diagrams show a summary of 11 independent experiments with 8 recipient mice per time point. Statistics were done with a two-tailed *t* test. $^{**}$ = p<0.01.

specific GM-CSF deficiency fail to secrete protective IgM and therefore develop strong pneumonia [13]. Our cell transfer experiments suggest not so much a lower homing of B cells to the lung, but rather that an impaired survival of B cells in the lungs of CD22-/- mice is responsible for this phenotype. CD22-deficient B cells show impaired survival *in vitro* and *in vivo* [9, 23], which may directly explain this lung phenotype.

In humans, attempts to find genetic factors contributing to susceptibility to pneumococcal disease so far have used candidate gene approaches. SNPs leading to a loss of the human CD22 protein have not been described so far. Systematic GWAS research on pneumococcal pneumonia susceptibility are lacking in the human, due to the difficulties in obtaining sufficient numbers of defined human subjects [24–26] and to the complex trait in susceptibility to infection, which is modulated by the individual's environment, age and gender [27]. Here we have described that the *cd22* locus as a novel strong susceptibility locus for pneumococcal disease in mice. The human *cd22* locus appears not to have been investigated in candidate gene approaches and before now the *Cd22* gene would not be included in a typical candidate gene search for genetic factors in pneumococcal disease. The data presented here provide strong evidence that it should be a candidate gene for future studies on human pneumonia.

## Materials and methods

### Ethics statement

All animal experiments done at the University of Leicester were conducted in strict accordance with U.K. Home Office licence PPL60/4327. The University of Leicester Ethical Committee and the U.K. Home Office approved the experimental protocols. Infected animals were closely monitored for disease signs and mice were humanely culled immediately they reached the 'lethargic' endpoint, or at the end of the experiment. Where indicated in the text, some infection experiments were done at the University of Erlangen in accordance with the licence of the Regierung of Unterfranken.

### Mice

Female BALB/cOlaHsd (abbreviated to BALB/c) and CBA/CaOlaHsd (CBA/Ca) were purchased from Harlan Laboratories (Bicester, UK) and were acclimatised for one week prior to use. Male and female CD22-deficient mice (Cd22tm1Lam) (on a pure C57BL/6 background) and control C57BL/6J mice (set up from littermates of CD22-deficient mice) [9] were obtained from the University of Erlangen and then bred at the University of Leicester. Mice used for infection experiments were between 9 and 18 weeks of age and were age-matched.

### Bacteria

*Streptococcus pneumoniae* strain D39 (NCTC 7466), a mouse virulent serotype 2 strain, was used for infection studies. Bacteria were identified as pneumococci by the Gram stain, catalase test, α-haemolysis on blood agar and by optochin sensitivity. The presence of capsule was

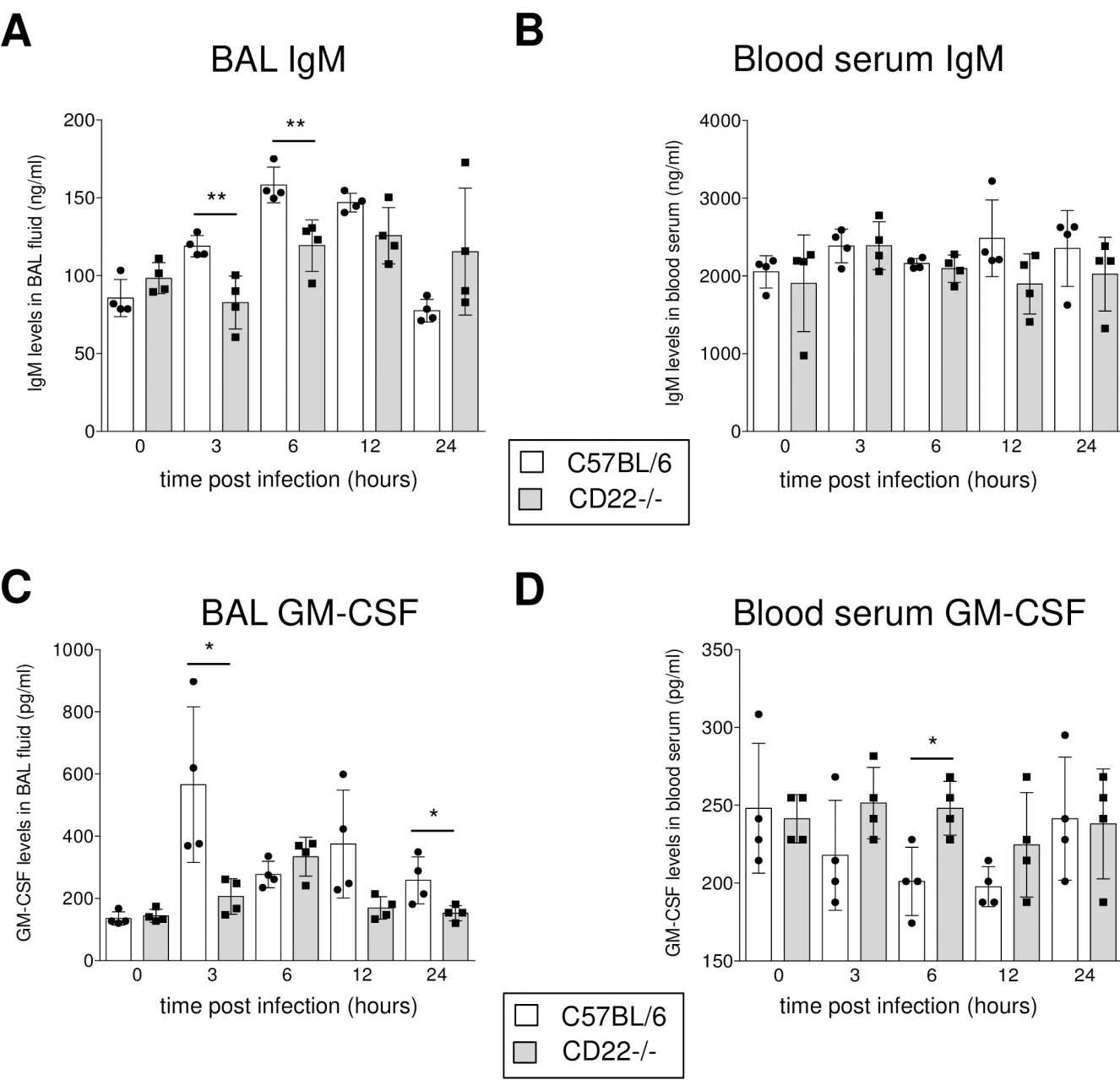

**Fig 6. IgM and GM-CSF levels in the BAL fluid of CD22-deficient mice after *S. pneumoniae* infections.** IgM (A, C) and GM-CSF (B, D) levels were measured by ELISA in BAL fluid of the lung (A, B) and in blood serum (C, D) after intranasal *S. pneumoniae* infection. Data are the means ± SEM. The p-values (* = p<0.05, ** = p<0.01, *** = p<0.001) were obtained with two-tailed *t*-test. Significant increase of IgM levels in the BAL fluid in control mice between 0 and 6 hours post-infection (p<0.01), but no significant change in IgM levels in the BAL fluid of CD22-/- mice (p>0.05). No significant changes were observed over the time course of the infection in the blood serum IgM levels of test and control groups (p>0.05). In the analysis of GM-CSF in the BAL fluid, it was observed a significant increase between 0 and 6 hours post-infection in control mice (p<0.05), but not in the CD22 knock-out mice (p>0.05). Over the time course of infection, the GM-CSF levels have significantly increased in the blood serum of control mice (p<0.05), although no significant change in the GM-CSF levels of CD22-/- mice (p>0.05).

confirmed via the Quellung reaction. Before use in infection studies, bacteria were passaged through mice to standardise the inoculum and were stored at -80˚C. When required, suspensions were thawed at room temperature and the inoculum prepared as described previously [28]. *Escherichia coli* FDA strain Seattle 1946 (ATCC 25922) was used for intratracheal infection.

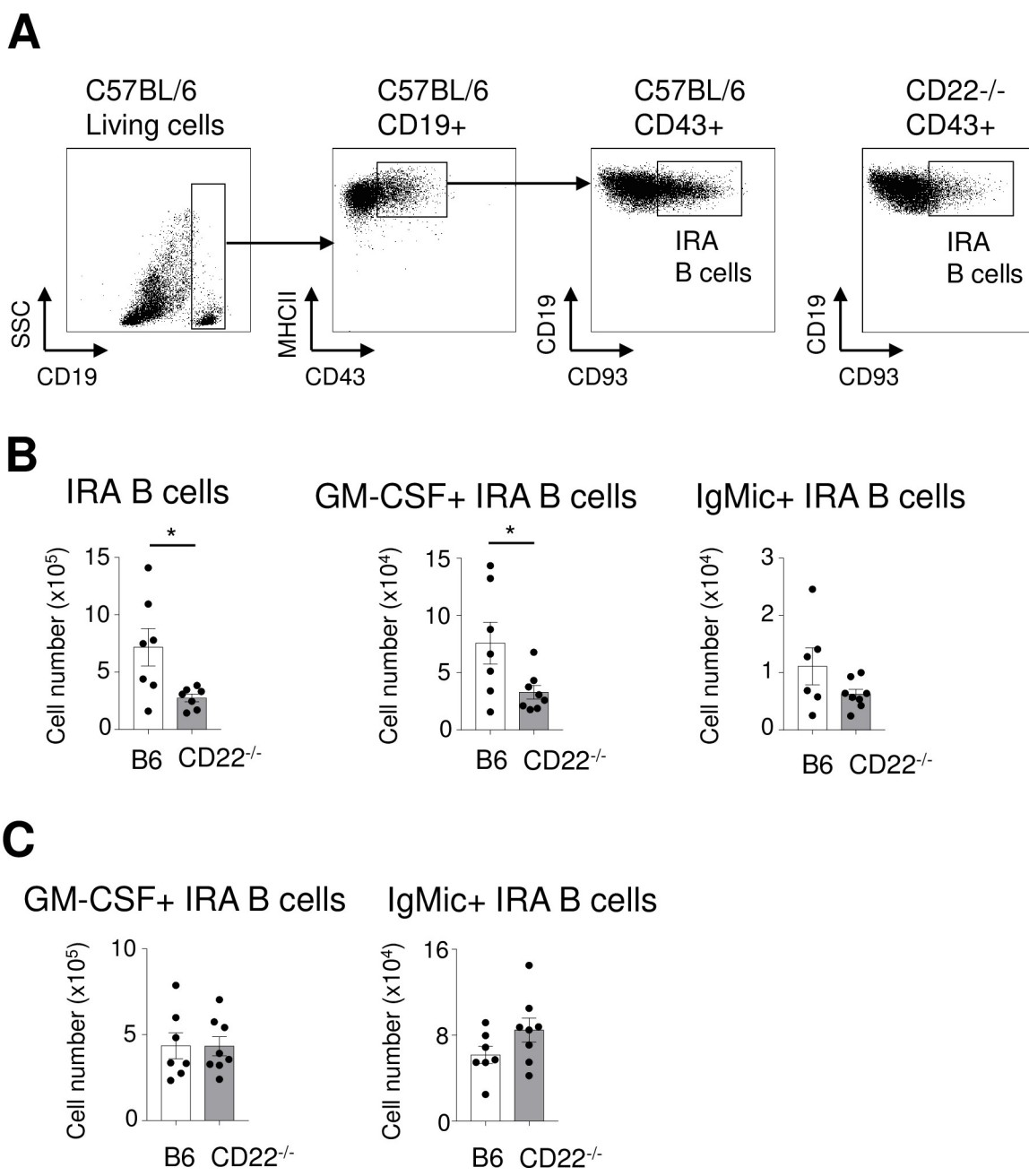

**Fig 7. CD22-deficient mice have a lower number of GM-CSF producing IRA B cells in the lungs after pneumococcal infections.**
C57BL/6 control and CD22-/- mice (C57BL/6 background) were infected with 1 x 10⁵ CFU *S. pneumoniae* intranasally and were analysed 72 hours after infection. (A) IRA B cells (MHCII+, CD19+, CD43+, CD93+) were identified in lung and spleen, as indicated for lung. (B) IRA B cells and GM-CSF and intracellular IgM+ (IgMic+) B cells were quantified in the lung. (C) GM-CSF+ and intracellular IgM+ (IgMic+) B cells were quantified and in the spleen. For B) and C) total cell numbers were calculated as percentage of total leukocyte numbers. * = p<0.05. n = 7 (C57BL/6) or n = 8 (CD22-/-) mice were used. A representative of two experiments is shown.

### Infection of mice

For intranasal infection, animals were lightly anaesthetised with 3% (v/v) isoflurane over oxygen, and an inoculum of 50μl containing 1x10⁶ CFU in PBS was administered drop by drop into the nostrils. Intranasal dosing of C57BL/6 strain, with an inoculum of 50μl containing

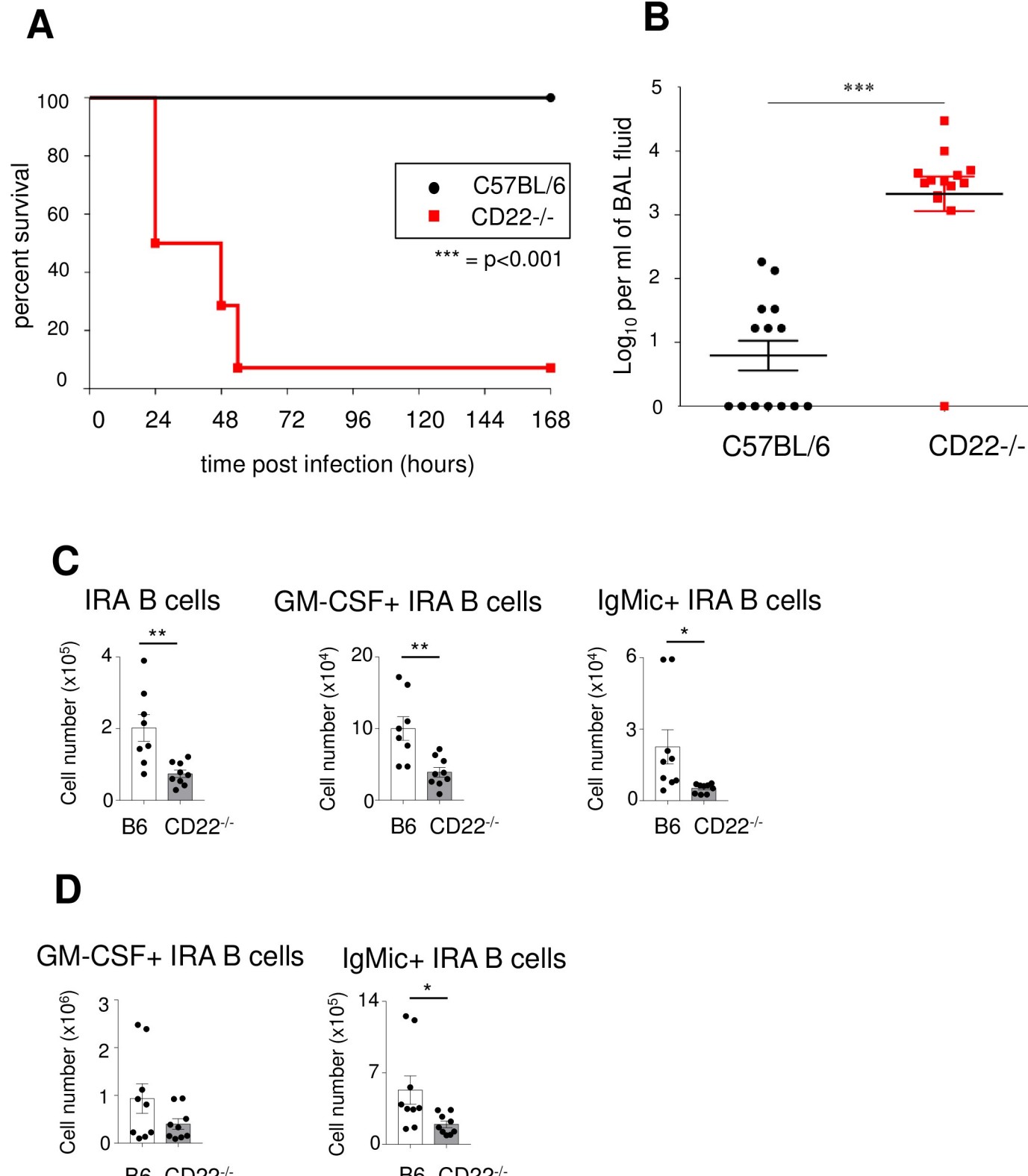

**Fig 8. CD22-deficient mice show a high susceptibility to *E. coli* intranasal infection and have a lower number of GM-CSF producing IRA B cells in the lungs.** Mice were infected with *E. coli* intranasally (A, B) or intratracheally (C-D). (A) Survival of intranasally infected mice was monitored for 7 days post-infection. (B) Number of bacteria per ml of BAL fluid at time of death. (C-D) 72 hours after *E. coli* infection, IRA B cells were identified in (C) lung and (D)

spleen, similarly as shown in Fig 7A. Within this IRA B cell population, GM-CSF+ and intracellular IgM+ (IgMic+) B cells were quantified in the lung (C) and in the spleen (D). For C) and D) total cell numbers were calculated as percentage of total leukocyte numbers. * = p<0.05, ** = p<0.01, *** = p<0.001. One representative of two experiments is shown.

$1 \times 10^6$ CFU of *S. pneumoniae* strain D39 (serotype 2, NCTC 7466) in PBS. For experiment shown in Fig 7, a dose of $1 \times 10^5$ CFU of *S. pneumoniae* in PBS was used for intranasal infection. For intranasal infection of mice with *E. coli*, $5 \times 10^6$ CFU in 50µl PBS was administered.

After infection, mice were frequently monitored and scored for visible signs of disease for 7 days (168 hours) according to the scheme of Morton and Griffiths [29]. The time to reach the lethargic state was the experimental endpoint and was designated as "survival time". Animals that were alive after 7 days post-infection were considered to have survived the pneumococcal infection.

In experiments to determine survival, on reaching the endpoint or after 7 days, mice were deeply anaesthetized with 5% (v/v) isoflurane over oxygen, and then culled by cervical dislocation. In time course experiments, pre-determined groups of animals were culled, as above, at predetermined times. After confirmation of death, lung and spleen tissues were removed and cell suspensions prepared for bacterial counting or flow cytometry. Blood samples were taken by cardiac puncture under terminal anesthesia or by tail-bleeding to determine the number of bacteria and immune cells in the blood.

For intratracheal infection, mice were anesthetized with isoflurane and infected intratracheally with $5 \times 10^6$ CFU *Escherichia coli* in a volume of 50 µl PBS with a 0.3ml syringe. Intratracheal *E. coli* infection of mice was done at the University of Erlangen.

## Preparation of tissue homogenates

Lung and spleen tissue were transferred into 10ml and 5ml, respectively, of sterile PBS, weighed, and then homogenised with an Ultra Turrax blender (Ika-Werke, Germany). Bacterial viable counts in homogenates were determined, followed by cell suspension preparation for flow cytometer analysis.

## Determination of bacterial numbers in the tissues

The viable count of bacteria in blood, lungs and spleen were determined at the pre-chosen times after intranasal infection. Viable counts were determined by serial dilution in sterile PBS and plating on blood agar plates containing 5% (v/v) defibrinated horse blood (Oxoid). Plates were incubated in 5% (v/v) $CO_2$ at 37˚C overnight and then bacterial colony numbers counted the following day.

## Flow cytometry

Cell suspensions were flushed through a 40µm cell strainer (BD Falcon) to remove tissue debris. Samples were centrifuged at 300 *g* for 5 min, at 4˚C, discarding the supernatant. The cell pellet was resuspended in ACK lysis buffer (Invitrogen) and incubated for 5 minutes, at room temperature, for lysis of erythrocytes. Once the incubation step finished, cell suspension was quenched with 10 ml PBS + 3% (v/v) FCS and spun down at 300 *g* for 5 min, at 4˚C. Mouse lung and spleen tissue cell pellets were resuspended with 1 ml PBS + 3% (v/v) FCS. Cell suspensions, at $1 \times 10^8$ cells/ml, were incubated with anti-mouse CD16/CD32 blocking antibody (clone 2.4G2, BD Pharmingen) before addition of the specific antibodies. Cell surface markers were stained using a combination of fluorescein isothiocyanate (FITC)-, phycoerythrin (PE)-, PE-Cy7- and allophycocyanin (APC)-conjugated monoclonal antibodies. Extracellular staining with monoclonal antibodies for CD4 (clone GK1.5, Biolegend), CD19 (clone

6D5, Biolegend), CD22 (clones OX-97 and Cy34.1, from Biolegend and BD, respectively), CD45 (clone 30-F11, Biolegend) and IgM (clone RMM-1, Biolegend) was performed according to the manufacturer's instructions. The presence of B-cell CD22 receptor in lung and spleen cells was assessed by staining with two monoclonal antibodies that bind to two different sites within domain two of the protein: clone OX-97 (Biolegend) and clone Cy34.1 (Becton Dickinson). In each experiment the appropriate isotype control monoclonal antibody and single staining controls were also included. Samples were analysed using a Becton Dickinson FACScalibur flow cytometer running CellQuest acquisition and analysed using FlowJo software (version 8.8.7, Tree Star).

## Extraction and sequencing of genomic DNA

DNA from BALB/c and CBA/Ca mouse strains were sequenced for polymorphisms flanking the *Spir1* QTL [2]. Firstly, DNA was extracted from tail tissue of mice, using a Nucleon kit (Gen-probe). Sequencing was then done at the Wellcome Trust Centre for Human Genetics by the High Throughput Genomics Group (UK), generating 8- to 10-fold coverage of a single lane of PE100nt per sample on the HiSeq2000. Paired-end Illumina reads from BALB/c and CBA/Ca mice were individually aligned to the reference genome (C57BL/6J, NCBIM37) using BWA. Detection of SNPs in each alignment was made using the Genome Analysis Toolkit (GATK) with default parameters. A filtering strategy was performed to identify high-confidence SNPs and reduce false positives. SNP sites that failed GATK's internal status check, quality <300 or had an allele ratio <0.8 were removed from further analysis. SNPs between CBA/Ca and BALB/c were then checked against the precompiled SNPs found in the 17 strains from the Mouse Genome Project (http://www.sanger.ac.uk).

## Primer design

Primers were designed for the *cd22* SNP identified from the sequencing data. Primers for pyrosequencing were designed using the PSQ Assay Design software (Biotage AB). Sets of three primers for each SNP were designed: one pair of primers for the PCR (one of which was biotinylated) and a sequencing primer for the pyrosequencing reaction. The minimum and maximum Tm for the PCR primers were 64 to 66˚C. The sequencing primers were designed with a maximum distance of three bases from the SNP. Primers were manufactured by MWG-Biotech.

## Extraction of genomic DNA and PCR

For pyrosequencing of DNA from BALB/c and CBA/Ca mice, DNA was extracted from mouse-tail tissue, using the DirectPCR lysis reagent (Bioquote Limited). 180 μl DirectPCR lysis and 20 μl proteinase K solution (20 mg/ml) was added to each tail sample and incubated overnight at 55˚C. After digestion, the samples were heated to 85˚C for 45 min and centrifuged for 10 seconds. 10 μl PCR reactions were then set up using 5 μl Taq PCR master mix (Qiagen), 0.2 μl forward primer and 0.2 μl reverse primer (at 10 pmol/μl), 2.6 μl nuclease free water and 2 μl DNA (~5 ng/μl). PCR reactions were run using the PCR program: 95˚C for 5 min, followed by 45 cycles of 95˚C for 15 sec, 60˚C for 30 sec and 72˚C for 15 sec. The final extension step was 5 min at 72˚C.

## Pyrosequencing

10 μl PCR product, 2 μl streptavidin-Sepharose beads (GE Healthcare), 38 μl binding buffer (Biotage) and 30 μl H$_2$O were combined in each well of a 96-well plate and mixed vigorously on a plate shaker for 5 min. The PCR products were then prepared using a vacuum work table

(Biotage AB). The biotinylated PCR products, attached to the filter probes of the vacuum apparatus, were immersed in 70% (v/v) ethanol for 5 seconds, denatured in PyroMark Denaturation solution (Biotage AB) for 5 sec (allowing only the biotin labeled strand of the PCR product to remain attached to the filter probes) and immersed in 1X PyroMark Wash buffer (Biotage AB) for 5 sec. The single-stranded PCR products were then re-suspended in a PSQ HS 96-well plate containing 0.5 μl sequencing primer (at 10 pmol/μl) and 11.5 μl annealing buffer (Biotage AB) per well. The plate was incubated at 80˚C for 2 minutes to allow the sequencing primer to anneal to the single-stranded PCR product. The PSQ 96-well plate and a PSQ HS 96 capillary dispensing tip holder (Biotage AB) containing enzyme, substrate and dNTPs (PyroGold reagent kit Biotage AB), were placed into a PSQ HS 96 Pyrosequencer (Biotage AB). The data were analysed using the SNP software (Biotage AB).

## Quantification of GM-CSF and IgM-producing B cells

72h after intratracheal infection with *E. coli*, organs were harvested and single cell suspensions obtained. For this, lungs were cut in small pieces and digested with 450 U/ml collagenase I (Sigma Aldrich), 125 U/ml collagenese IX (Sigma Aldrich), 60 U/ml hyaluronidase (Sigma Aldrich), 60 U/ml Dnase (Sigma Aldrich) in 20 mM Hepes pH7.4 (Thermo Scientific) for 1 hour at 37˚C, while shaking. Spleens then were homogenised through a 40 μm nylon mesh. Total viable cell numbers were determined using trypan blue (Carl Roth). Lung homogenates and splenocytes were stained for cell surface markers for 20 minutes at 4˚C, fixed for 20 minutes (BD Cytofix, BD Bioscience) at 4˚C, washed and permeabilized for 20 minutes (BD Phosflow Perm Buffer III, BD Bioscience) at 4˚C. Then cells were washed before staining with anti-GM-CSF and anti-IgM antibodies for 20 min at 4˚C and washing twice. The following antibodies were used for flow cytometric analyses: anti-CD43-FITC (BD Biosciences), anti-CD93-BUV395 (BD Biosciences), anti-IgM-BV650 (BD Biosciences), anti-MHCII-BV711 (BD Biosciences), anti-CD45.2-BV786 (BD Biosciences), anti-CD19-BV421 (BD Biosciences), anti-GMCSF-PE (BD Biosciences), anti-IgM-PerCP Cy5.5 (BD Biosciences), IgG2a-PE (BD Biosciences), IgG2a-PerCP Cy5.5 (BD Biosciences). B cell populations were identified, as described previously [30]. Data were acquired on a Celesta (BD Biosciences) flow cytometer and analyzed with FlowJo 10 (FlowJo LLC).

## Cell transfer assay

WT (C57BL/6)and CD22-/- splenocytes were isolated and labelled with CTV or CFSE (Molecular Probes, 1:1000 dilution in PBS, labelling for 10 minutes at room temperature), respectively. In half of the experiments splenocytes of the two strains were labelled after swapping the two dyes to exclude any effects of the dyes. Equal numbers of labelled donor cells ($15–30x10^6$ cells per genotype) were mixed 1:1 and injected intravenously into C57BL/6 WT recipients. After 2, 24 and 48 hours lungs and spleens of the recipient mice were isolated, lungs were digested with Collagenase D (50mg/ml) / DNAse I (50mg/ml) for 30 minutes at 37˚C, single cell suspensions of the lungs and spleens were produced and erythrocyte lysis was performed. Lung and spleen cells were stained for flow cytometry with anti-B220 and anti-CD3 antibodies (Biolegend) to identify B220[+] B cells and CD3[+] T cells. Draq7 dye was used for live/dead staining. The proportion of WT and CD22[-/-] donor cells was evaluated by calculation of the ratio: percentage of CTV or CFSE-positive B or T cells found in the lung or spleen divided by the percentage of CTV or CFSE-positive B or T cells injected (input) x 100.

## Statistics

Data were analysed using a two-tailed unpaired Student *t* test or one- or two-way ANOVA, as appropriate. Survival data were analysed using Kaplan-Meier survival analysis. Results with p-

values less than 0.05 were considered significant. Statistical analysis was performed with the aid of Graph Pad Prism (version 5.0b).

## Supporting information

**S1 Fig. The stop codon identified in the *cd22* gene of CBA/Ca does not occur in 37 inbred mouse strains from Jackson Laboratories.** The SNP found in the *cd22* gene in the CBA/CaO-laHsd inbred strain is located at position 7:30,877,586 (chromosome position highlighted in light blue), and is absent in all other sequenced strains. The SNPs in the figure have the following consequences: T (dark red) stop gained; A, C, G and T (light red), NMD transcript variant; A, G and T, (yellow) missense/initiator codon variant; A and C (green), synonymous/stop retained variant. The asterisk symbol on the SNP indicate that the SNP was observed in multiple sequences. Exon 4 SNPs (blue highlighted box with an arrow indicating the reverse orientation) are located between SNP positions 7:30,877,522–30,877,839. Data were collected from the Wellcome Trust's Sanger Institute website (http://www.sanger.ac.uk).
(PDF)

**S2 Fig. B cells in CBA/Ca mice do not express CD22 as revealed by staining with Cy34.1 antibody.** Flow cytometric analysis of splenocytes and lung cells from BALB/c and CBA/Ca mice stained for CD19 and CD22 (MAb Cy34.1) 24 h after intranasal infection with *S. pneumoniae*.
(PDF)

**S3 Fig. B cell numbers in the lungs and spleen of CBA/Ca and BALB/c mice after pneumococcal infection.** Mice were intranasally infected with 1 x $10^6$ CFU and organs were collected at 0, 6, 12 and 24 hours post-infection. B cell numbers were determined by gating IgM-/+ CD19+ cells in lung (A) and spleen (B) by flow cytometry. Significant increase of B cells numbers over time ($p<0.01$, from 0 to 24 hours p.i.) in the lung tissue of BALB/c mice (Figure A) and significant decrease in the spleen of BALB/c mice ($p<0.01$). Examples of the 24 hours time point are shown in the top panels. Data are representative of two independent experiments with $> 4$ mice per group. The p-values (** = $p<0.01$) were obtained with two-way ANOVA.
(PDF)

**S4 Fig. Slightly impaired IgM levels at 24 hours in BAL fluid of CBA/Ca mice after pneumococcal infection.** Mice were intranasally infected with $10^6$ CFU and BAL fluid was collected at 0, 6, 12 and 24 hours post-infection. Total IgM and IgG levels were determined by ELISA. Data are representative of two independent experiments with $> 4$ mice per group. ** $p< 0.01$ (two-way ANOVA).
(PDF)

## Acknowledgments

We thank members of the GEMS core at MRC Harwell for their excellent technical support and for their help with Pyrosequencing. We thank Dr. A. Gessner and Dr. J. Glaesner, University of Regensburg, for help with *S. pneumonia* infections. We thank C. Koller for technical support.

## Author Contributions

**Conceptualization:** Vitor E. Fernandes, Georg F. Weber, Lars Nitschke, Peter W. Andrew.

**Data curation:** Giuseppe Ercoli, Alan Bénard, Carolin Brandl, Hannah Fahnenstiel, Jennifer Müller-Winkler.

**Formal analysis:** Vitor E. Fernandes, Giuseppe Ercoli, Alan Bénard, Paul Denny.

**Funding acquisition:** Georg F. Weber, Lars Nitschke, Peter W. Andrew.

**Project administration:** Lars Nitschke, Peter W. Andrew.

**Supervision:** Georg F. Weber, Lars Nitschke, Peter W. Andrew.

**Writing – original draft:** Vitor E. Fernandes, Lars Nitschke, Peter W. Andrew.

**Writing – review & editing:** Vitor E. Fernandes, Georg F. Weber, Lars Nitschke, Peter W. Andrew.

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
