## [Decision Letter · Decision Letter 0]

15 Aug 2019

Dear Dr. Nitschke,

Thank you very much for submitting your manuscript "The B-cell inhibitory receptor CD22 is a major factor in host resistance to Streptococcus pneumoniae infection" (PPATHOGENS-D-19-01231) for review by PLOS Pathogens. Your manuscript was fully evaluated at the editorial level and by independent peer reviewers. The reviewers appreciated the attention to an important problem, but raised some substantial concerns about the manuscript as it currently stands. These issues must be addressed before we would be willing to consider a revised version of your study. We cannot, of course, promise publication at that time.

We therefore ask you to modify the manuscript according to the review recommendations before we can consider your manuscript for acceptance. Your revisions should address the specific points made by each reviewer.

(1) A letter containing a detailed list of your responses to the review comments and a description of the changes you have made in the manuscript. Please note while forming your response, if your article is accepted, you may have the opportunity to make the peer review history publicly available. The record will include editor decision letters (with reviews) and your responses to reviewer comments. If eligible, we will contact you to opt in or out.

(2) Two versions of the manuscript: one with either highlights or tracked changes denoting where the text has been changed; the other a clean version (uploaded as the manuscript file).

Additionally, to enhance the reproducibility of your results, PLOS recommends that you deposit your laboratory protocols in protocols.io, where a protocol can be assigned its own identifier (DOI) such that it can be cited independently in the future. For instructions see http://journals.plos.org/plospathogens/s/submission-guidelines#loc-materials-and-methods

We hope to receive your revised manuscript within 60 days. If you anticipate any delay in its return, we ask that you let us know the expected resubmission date by replying to this email. Revised manuscripts received beyond 60 days may require evaluation and peer review similar to that applied to newly submitted manuscripts.

[LINK]

Sincerely,

Dana J. Philpott

Associate Editor

PLOS Pathogens

Michael Wessels

Section Editor

PLOS Pathogens

Kasturi Haldar

Editor-in-Chief

PLOS Pathogens

orcid.org/0000-0001-5065-158X

Grant McFadden

Editor-in-Chief

PLOS Pathogens

orcid.org/0000-0002-2556-3526

The reviewers were overall positive about the findings. One important aspect that should be addressed is to provide more convincing evidence that it is indeed B cell recruitment rather than in situ proliferation that is responsible for the phenotype. Moreover, the manuscript would be strengthened with data showing that in the pneumococcal model, there are also fewer IRA B cells (rather than just with the E coli infection).

Reviewer's Responses to Questions

**Part I - Summary**

Reviewer #1: The authors present convincing evidence that a SNP resulting in loss of function of CD22 largely accounts for the increased susceptibility of CBA/Ca mice to pneumococcal lung infection, relative to other inbred mouse strains. Moreover, CD22-/- mice in the C57BL/6 background exhibited enhances susceptibility relative to the wild type. Clues as to the underlying mechanism were also provided by data showing a defect in recruitment of IRA B cells, resulting in lower levels of GM-CSF and IgM. This study provides new information on the hitherto unrecognised role of CD22 in host resistance to bacterial infection and paves the way for further molecular epidemiological/GWAS studies, including in human populations. For this reason, it is an important contribution.

Reviewer #2: In this manuscript, the authors describe an important and previously unrecognized role for CD22 in susceptibility of infection to Streptococcus pneumoniae. This works stems from an earlier report by the authors on a loci within CBA/Ca mice that causes high susceptibility to infection. This loci contains CD22, an important B cell inhibitory co-receptor, and it is found that a premature stop codon is located in the Cd22 gene within this strain that causes no functional CD22 to be expressed on their surface. To test if this loss of functional CD22 in CBA/Ca mice is responsible, the author carry out studies in C57BL/6 mice on a CD22+/+ and CD22-/- background. Studies in C57BL/6 mice support the authors’ claim that CD22 is directly involved in susceptibility Streptococcus pneumoniae infection. The authors search for mechanism is far from complete, but they do reveal a key differences in B cell recruitment to the lung, supporting earlier reports of B cells being an important immune cell limiting Streptococcus pneumoniae infection in mice. The results are interesting, with many key questions remaining, but should be published after the points below are addressed:

Reviewer #3: The paper by Fernandes et al., seeks to describe the role of CD22 in susceptibility to pneumococcal disease. The authors hypothesis is that a point mutation in CD22 in CBA mice is the cause of their susceptibility to pneumococcal disease. They base their argument on the fact that CD22 KO mice are also susceptible to pneumococcal infection. This is an interesting hypothesis, however the main issue is that the data supporting this comes from the CD22 KO which is in the C57 background. Therefore, it is difficult to see how the authors can prove that CD22 really is the factor leading to susceptibility in CBA mice. Yes, there is an association i.e. it seems that in C57 mice lack of CD22 does lead to increased susceptibility to pneumococcal infection, but we need to see a direct link/proof in CBA mice. This is not provided, which is surprising as there are a number of experiments which could have been done to address this.

The only data provided from the CBA mouse background is the supplementary data which shows total B cells in lung and spleen tissue from CBA vs Balb/c mice. What is interesting is that there are no differences in B cells between these two strains of mice in lung or spleen, with the exception of 24hrs in lung B cells in CBA mice, but by this time the CBA mice already have high CFU levels in their blood based on previous papers and are dying from their infection. Indeed, CBA mice develop bacteremia as early as 6hrs post infection, hence the essential role of B cells in this mouse strain is not really supported.

See comments in additional exps that may strenghten paper.

**Part II – Major Issues: Key Experiments Required for Acceptance**

Reviewer #1: Page 8. Whilst the difference between C57BL/6 and CD22-/- mice in numbers IRA B cells in lungs after challenge with E. coli are convincing, it is a little puzzling why the authors did not attempt the same experiments in mice challenged with D39. It would be preferable if such data were included in the manuscript.

Reviewer #2: (No Response)

Reviewer #3: [1] It would be good to see IgM and IgG levels (including subsets) in CBA vs Balb/c mice (lungs and blood) following infection with pneumococci at the timepoints used (0, 6, 12, 24 etc).

[2] It would be good to see a more detailed B cell analysis, including subsets (B1, B2 etc) in above groups/times in lungs.

[3] Would be interesting to see adoptive transfer of B1 cells into CBA mice. If their susceptibility is due to a lack of these cells, then adoptive transfer would answer the question of whether this is true or not.

**Part III – Minor Issues: Editorial and Data Presentation Modifications**

Reviewer #1: Line 84. pneumoniae.

Line 92. LOD needs to be defined.

Line 93. The definition of QTL needs to be here rather than on line 99.

Line 176. It is not strictly correct to say there were no bacteria in the blood of infected C57BL/6 mice, as there was 1 culture-positive mouse at 12 h.

Line 187. should be Suppl Fig 2B.

Line 211. GM-CSF in blood was actually significantly higher in CD22-/- mice at 6 h, so perhaps this sentence needs to be qualified.

Fig. 4. The A, B, C and D labels are missing from the panels.

Reviewer #2: 1. No direct evidence is provided to demonstrate a defect in B cell recruitment or strictly that it is CD22 expression on B cells that is the key factor. Accordingly, I have 4 inquiries:

a) Can the authors rule out impaired proliferation of CD22-/- resident lung B cells?

b) Is it known that B cells home to the lung in a similar manner as the Peyer’s patches? Are HEVs involved? If so, it would certainly be worth discussing this similarity more directly. If not, it should be stated that a previously undiscovered homing mechanism would necessarily need to be at play.

c) Can the authors rule out a role for DCs given that CD22 is known to be expressed on these cell types? Is there any known link between DCs and S. pneumoniae infection? In this respect, it is worth pointing out that Ma et al. (2012, J Virol) determined that loss of CD22 results in a significant decrease in several cytokines which was due to loss of CD22 in cDC2.

d) Does S. pneumoniae express ligands of CD22 that could be contributing to these effects? Have the authors (or anyone previously) looked at staining of S. pneumoniae with CD22-Fc?

2. For the study in Ref19, it would be worth discussing any possible differences in: a) route of administration, b) strain background, c) primary site of infectivity between Staphylococcal and Streptococcal infection between studies.

3. Please provide suitable ref for the statement on Lines 146-148 about antibody specificity.

4. Lines 224-226: Data not shown?

5. Lines 228-237: Assume this is referring to intranasal infection? Since the experiment described directly above is intratracheal, it should be clearly stated.

6. Are there any reported SNPs in humans that result in loss of CD22 protein or loss of functional CD22? Worth mentioning either way in the discussion.

7. Typos:

Line 222: Type: C22

Line171: CD22 -/- to CD22-/-

Reviewer #3: (No Response)

PLOS authors have the option to publish the peer review history of their article (what does this mean?). If published, this will include your full peer review and any attached files.

Reviewer #1: No

Reviewer #2: No

Reviewer #3: No

---

## [Decision Letter · Decision Letter 1]

15 Feb 2020

Dear Dr. Nitschke,

Thank you very much for submitting your manuscript "The B-cell inhibitory receptor CD22 is a major factor in host resistance to Streptococcus pneumoniae infection" for consideration at PLOS Pathogens. As with all papers reviewed by the journal, your manuscript was reviewed by members of the editorial board and by several independent reviewers. The reviewers appreciated the attention to an important topic. Based on the reviews, we are likely to accept this manuscript for publication, providing that you modify the manuscript according to the review recommendations.

Thank you for submitting your work to PLOS Pathogens. The reviewers were mostly satisfied with the revision; there were some minor points to be addressed outlined by Reviewer 2. Please make these changes with respect to data presentation (individual data points) as well as addressing the comments highlighted by Reviewer in the text of the paper.

Sincerely,

Dana J. Philpott

Associate Editor

PLOS Pathogens

Michael Wessels

Section Editor

PLOS Pathogens

Kasturi Haldar

Editor-in-Chief

PLOS Pathogens

orcid.org/0000-0001-5065-158X

Michael Malim

Editor-in-Chief

PLOS Pathogens

orcid.org/0000-0002-7699-2064

Thank you for submitting your work to PLOS Pathogens. The reviewers were mostly satisfied with the revision; there were some minor points to be addressed outlined by Reviewer 2. Please make these changes with respect to data presentation (individual data points) as well as addressing the comments highlighted by Reviewer in the text of the paper.

Reviewer Comments (if any, and for reference):

Reviewer's Responses to Questions

**Part I - Summary**

Reviewer #1: The authors have satisfactorily addressed the issues raised in my original review.

Reviewer #2: The issues brought up by the reviewers have mostly be addressed in a satisfactory manner. My final comments more about presentation and description of the data.

**Part II – Major Issues: Key Experiments Required for Acceptance**

Reviewer #1: N/A

Reviewer #2: (No Response)

**Part III – Minor Issues: Editorial and Data Presentation Modifications**

Reviewer #1: N/A

Reviewer #2: 1) It would strengthen the presentation to show individual data points in the bar graphs.

2) i) In the new adoptive transfer data, the authors describe it as 'migration'. While I am admittedly not an expect in this area, I think migration refers more to movement towards a signal, which may be more appropriate during infection. Since these particular experiments were carried out in healthy mice, I believe homing is the more correct term.

ii) They describe infer that the adoptive transfer experiment provides insight into 'proliferation in the lung'. I do not see evidence that they examined this aspect (which could have been done in the context of fluorescence dilution). It appears that they have only investigated homing and retention. Wording to describe this in the text should be modified appropriately.

3) lines 46 and 113- it is written 'a lack of B cells' in CD22-null mice post infection. While B cell population in the lungs CBA/Ca mice is significantly reduced after bacterial inoculation, there are a few that were observed. I think it is an overstatement to say that these mice lack the B cells.

4) In Figures 7B/7C and 8C/8D, the authors should report if these are normalized (e.g. per mg tissue).

PLOS authors have the option to publish the peer review history of their article (what does this mean?). If published, this will include your full peer review and any attached files.

Reviewer #1: No

Reviewer #2: No
---

## [Editor Report · Decision Letter 2]

6 Mar 2020

Dear Dr. Nitschke,

We are pleased to inform you that your manuscript 'The B-cell inhibitory receptor CD22 is a major factor in host resistance to Streptococcus pneumoniae infection' has been provisionally accepted for publication in PLOS Pathogens.

Best regards,

Dana J. Philpott

Associate Editor

PLOS Pathogens

Michael Wessels

Section Editor

PLOS Pathogens

Kasturi Haldar

Editor-in-Chief

PLOS Pathogens

orcid.org/0000-0001-5065-158X

Michael Malim

Editor-in-Chief

PLOS Pathogens

orcid.org/0000-0002-7699-2064

---

## [Editor Report · Acceptance letter]

8 Apr 2020

Dear Dr. Nitschke,

We are delighted to inform you that your manuscript, "The B-cell inhibitory receptor CD22 is a major factor in host resistance to *Streptococcus pneumoniae* infection," has been formally accepted for publication in PLOS Pathogens.

Best regards,

Kasturi Haldar

Editor-in-Chief

PLOS Pathogens

orcid.org/0000-0001-5065-158X

Michael Malim

Editor-in-Chief

PLOS Pathogens

orcid.org/0000-0002-7699-2064